# Community Engagement and the Effectiveness of Free-Roaming Cat Control Techniques: A Systematic Review

**DOI:** 10.3390/ani14030492

**Published:** 2024-02-02

**Authors:** Daniela Ramírez Riveros, César González-Lagos

**Affiliations:** 1Centro de Investigación en Recursos Naturales y Sustentabilidad (CIRENYS), Universidad Bernardo O’Higgins, Santiago 8370993, Chile; 2Center of Applied Ecology and Sustainability (CAPES), Santiago 8331150, Chile; cesar.gonzalez.l@uai.cl; 3Departamento de Ciencias, Facultad de Artes Liberales, Universidad Adolfo Ibáñez, Santiago 8320000, Chile

**Keywords:** *Felis catus*, feral cats, free-roaming cats, population management, cat management, community engagement

## Abstract

**Simple Summary:**

Free-roaming and feral cats, along with their impacts on ecosystems and humans, have been debated globally. Cat welfare, overpopulation, and environmental and public health problems have prompted interest in controlling their populations. Several techniques exist to control cat populations, but community engagement may not always be considered. A systematic literature review was conducted to evaluate if community engagement influences the effectiveness of control techniques in managing cat populations, excluding culling. The greatest control occurred with highest community engagement; adoption and education determined the effectiveness. While research on cat control exists, few studies evaluate community engagement and technique effectiveness. This information is particularly relevant in countries that explicitly incorporate certain control techniques into their legislation.

**Abstract:**

Although free-roaming and feral cat control techniques are often applied in human communities, community engagement is not always considered. A systematic literature review following an update of the Preferred Reporting Items for Systematic reviews and Meta-Analyses (PRISMA 2020) methodology was conducted to evaluate whether community engagement influences the effectiveness of control techniques, excluding culling, in managing cat populations. The degree of community engagement was estimated based on the number of roles reported during the application of the control technique, which included adoption, trapping, care, and/or education. Education followed by adoption was the determining factor in the decreasing cat populations over time. The limited evaluations of control technique effectiveness, narrow geographical scope, and our simple measure of engagement emphasize the need for more detailed studies. These studies should evaluate the effectiveness of control techniques, while considering community engagement more comprehensively.

## 1. Introduction

For 9000 years, domestic cats (*Felis silvestris catus*) have been companions to humans, stemming from the domestication of their ancestor, the African wild cat (*Felis silvestris lybica*) [1]. The exact process of this domestication remains uncertain, but evidence suggests that African wildcats essentially “self-domesticated” due to a mix of ecological and sociocultural factors [2]. Currently, the roles and presence of domestic cats in human societies and natural ecosystems remain contentious [3]. Cats uniquely embody a dual identity, being both wild (autonomous) and domestic (closely associated with and dependent on humans) [4]. Consequently, the extent of their interaction and impact on wildlife is closely tied to how felids are managed, particularly in terms of confinement [5]. Cats can be classified into two broad categories, based on human control. Firstly, “indoor” cats are under strict human control, pose minimal threats to wildlife, and their welfare depends on their owners [4]. Secondly, “free-roaming” cats (also known as “free-ranging”) are not confined to houses or other enclosures. These include owned and semi-owned cats that roam free some or all the time (i.e., stray cats), and feral cats [4,6,7]. “Feral cats” are free-roaming cats that live untamed. The cat was either abandoned and reverted to a more feral state or was born outdoors and had little or no human contact. Feral cats are afraid of people and avoid contact whenever they can [4,5,8,9,10]. Moderate or null human control of cats compromises their welfare and increases the likelihood of injuries and diseases. It should be noted that these types of cats can interact with each other in the same context and can quickly transgress the provided definitions, confounding attempts to categorize them [6].

A growing body of research has assessed the impact of free-roaming cats on wildlife. Islands, in particular, bear the brunt of the impact of cats, which cause vertebrate extinction. Feral cats, known to prey on introduced mammals, also attack native birds and reptiles [11,12]. It should be noted that there are multiple reasons for wanting to control cat populations, apart from the impact they generate on wildlife, but also the inconvenience they can cause to people (toileting, fighting, and hunting behaviors, for example) and the welfare of the cats themselves [4,6,7]. 

Various methods exist to manage the size of free-roaming cat populations. The choice of control method depends on many factors, but we will focus on two key factors: (i) whether the cats are owned or not, and (ii) the level of responsibility and control exercised by humans. Some of the best-known methods include: (1)**Surgical sterilization (SS)** is a method applicable to owned cats, where owners bring their cats to a center for surgical sterilization before taking them home.(2)**Trap–neuter–return (TNR)** involves trapping, sterilizing, and returning free-roaming cats to their original capture site [7]. Cats are trapped, vaccinated against rabies where that disease occurs, sterilized, ear-tipped (1 cm of the left ear removed to indicate that they are sterilized) and returned to their capture site [13]. There are some variations in TNR, such as **trap–test–vaccination–test–alteration–return (TTVAR)**; cats undergo the same process as in TNR, including testing and alteration without the explicit inclusion of a caretaker, but if they return to their colony, the caretakers provide food and monitor their welfare [14].(3)Trap–remove (TR) controls unwanted cat populations by trapping and removing cats from a specific area through adoption or euthanasia [7]. Variations may include placing cats in sanctuaries instead. A well-known variation is trap–euthanize (TE), where some trapped cats are returned to owners or adopted, but most unwanted excess cats are euthanized [15,16].

There are other lethal management methods, often termed “culling”, such as shooting, poisoning, and euthanizing entire cat populations. These methods raise ethical concerns regarding cat management, resulting in substantial controversy [17,18].

Permanent sterilization has no immediate impact on the population size of free-roaming cats, because they remain in the environment. Nevertheless, this method can exert a long-term effect by significantly reducing the proportion of the population capable of reproduction. According to Miller et al. [19], control programs for cat populations can be classified as either long-term or short-term, based on their time of application. Short-term control techniques, with a duration of less than five years, often yield varying levels of success in controlling cat populations. For example, a one-year TNR program in New York, studied by Kilgour et al. [20], showed no significant differences in the total population estimates before and after the program. On the other hand, Kennedy et al. [7] highlighted that long-term TNR programs aim to optimize the reduction or elimination of free-roaming cat populations, resulting in significant population declines in both rural and urban areas.

The success of control programs, as well as their effectiveness, depends not only on their duration or the target cat population but also on the socio-environmental context. This context encompasses the physical and social environments in which activities take place, including the cultural and institutional factors with which human groups interact [21]. Considering the socio-environmental context and involving local interests in strategy design, decision-making, and implementation are crucial factors in determining the outcome of the strategy [22]. Free-roaming cat management programs may require the collective support and participation of a diverse group of community members with different backgrounds, capabilities, opportunities, and motivations [23]. There are studies in other fields such as health promotion, health literacy, and wellness, in which community involvement enhances the empowerment of community members in decision-making [24]. However, evidence supporting the importance of a collaborative, community-based approach to cat population control is currently scarce [25]. In fact, several barriers (lack of social support networks, low sense of personal autonomy, and distrust towards control service providers) hinder people’s motivation to participate in the implementation of any specific technique, resulting in communities not being involved and not helping to curb the feline overpopulation problem [24]. This research aimed to assess how the type and quantity of community involvement influence the application of control techniques and, consequently, the effectiveness of reducing cat populations.

## 2. Materials and Methods

We conducted a systematic literature review following the Preferred Reporting Items for Systematic reviews and Meta-Analyses (PRISMA 2020) guidelines [26], which aid authors in transparently documenting the purpose, methodology, and findings of their systematic review. Google Scholar was used (Appendix A). Unlike databases such as Web of Science and Scopus, Google Scholar encompasses a wide range of sources, including scientific articles, conference papers, abstracts, technical reports, theses, dissertations, and academic books, which may be relevant to our research objectives. The following keyword combinations were used during the initial bibliographic search: (“free-ranging cat” OR “free-roaming cat” OR “feral cat” OR “stray cat”) AND (“trap-neuter-return” OR “trap-neuter-release” OR “trap-remove” OR “trap-euthanize” OR “trap-neuter-vaccinate-return”). We applied a temporal filter, considering articles from 1992 to 2023, as one of the earliest instances of feline control technique application dates back to 1992 [27]. We included articles published in both English and Spanish. Given the inconsistency in terminology for referring to cat types and control techniques [28], we included synonyms that were identified during the initial bibliographic search. Technique abbreviations were included as synonyms because they are widely used as official acronyms [7,13,29] (see Table A1).

For lethal techniques, we excluded the term “culling”, because it encompasses the euthanasia of entire populations, shooting, poisoning, and predator introduction, which raises significant ethical and welfare concerns [17]. Consequently, we focused on established techniques that can be applied in different contexts without generating substantial controversy, while ensuring the welfare of cats as far as possible (see also “Study Limitation”).

Following systematic review protocols in line with O’dea et al.’s recommendations [30], we selected studies from the bibliographic search that met our pre-established eligibility criteria (see Table 1). 

All articles that met criteria 1, 2, and 3 were included in a structured matrix for characterization and synthesis (see Appendix A). Studies consisting solely of simulations were excluded. Although these articles incorporate existing population and demographic data into models, the execution of the control strategy is theoretical and uses computer algorithms rather than experimentation. Appendix A lists simulation-based articles that were excluded.

The articles that met the four criteria were separated into a second matrix (Appendix A), and the information extracted from each article was indicated by the following variables:-**Type of management**: the cat population control techniques used (TNR and its variations, TR, TE);-**Type of cat**: type of cat involved (owned or un-owned stray cat, and feral cat);-Socio-environmental context, for which the following variables were considered:(i)**Type of human grouping**: urban, rural, or non-human grouping.(ii)**Geographical area** where control was applied.(iii)**Countries** where control was applied.-**Objectives**: the objective of the article that is related to evaluate the effectiveness of a control technique;-**Methodology**: methodology used to apply the control technique;-**Results**: main observations and results of applying the control technique;-**Conclusion**: conclusion of the author(s) regarding the application of the control technique;-**Years of application**: the length of time (years) during which the effectiveness of the control technique was evaluated;-**Initial population**: initial feline population to which the control technique was applied;-**Final population**: the resulting population at the end of the evaluation period of its effectiveness.

To identify the degree of **community participation**, the following variables were considered:-**Participation in human groupings:** as a minimum category, it was considered whether there was participation of the communities where the control method was applied, as indicated in the “Materials and Methods” and “Results” sections of each article.-**Type of participation of the human groups:** studies in which the participation of human groups was classified as one of the following:(i)**Adoption**: participation in adoption campaigns.(ii)**Trappers**: trapping cats from colonies.(iii)**Caretakers**: care of cat colonies.(iv)**Education**: education/awareness of control methods or impacts of feral/free-roaming cats.

This classification was derived by reading articles that met Criterion 3, using keywords (ultimately becoming the names of the categories) found within the articles.

The effectiveness of a control technique can be determined by the number of cats sterilized, adopted, litter prevented, etc., depending on the purpose of its application. In this study, we focused on population reduction over time for a given colony. To assess the latter, the difference between the population sizes before and after the application of the control was evaluated. To mitigate the differences in magnitude while preserving relative variation, we employed a logarithmic transformation of the efficacy variable. This transformation yields a statistically comparable numerical value [31]. Participation was also summarized into high (four roles), medium (three roles), and low (two and one role) categories according to the number of roles community members played in the application of the control technique. Graphical elements were created in R Studio [32], and EndNote 20 [33] was used to store and classify the elements during the review.

## 3. Results

A bibliographic search was conducted between 24 February and 18 April 2023. We obtained 985 references that were published between 1992 and 2023. These references were registered and classified based on the eligibility criteria (see Table 1). Subsequently, we repeated the search, incorporating the synonyms retrieved during the review of the original 985 articles (see Table A1). As a result of this second search, we obtained 1735 articles. Eleven duplicates were eliminated, leaving a total of 1724 articles.

For selection based on criteria 1 and 2, the titles and abstracts of 1724 articles were analyzed (Figure 1). To verify criterion 3, the 44 articles resulting from the previous process were organized in a matrix to analyze their objectives, methodology, results, and conclusions (Appendix A). Applying all the pre-established eligibility criteria, 1713 articles were discarded, leaving a total of 11 articles (Figure 1, Appendix A).

Of the eleven articles, seven pertained to cat population control initiatives in the United States (64%), followed by two in Italy (18%), one in Australia (9.1%), and one in Canada (9.1%). Ten of the eleven articles described initiatives in urban contexts, with one focusing on a rural context. All eleven articles employed the TNR technique, with ten using the free-roaming cat concept and only one focusing on feral cats. All articles reported community engagement in cat population control. This engagement was categorized into one or more of the four groups described in the analyzed articles. The categories and their descriptions are listed in Table 2.

Four of the 11 evaluated articles had the highest number of community participation roles (n = four). Participation was summarized into high (n = four studies), medium (n = three studies), and low (n = four studies) categories (Table 3). Because of the magnitude of the differences between the values of the population sizes subjected to the control techniques (range = 62–15,718 individuals), a logarithmic transformation was applied to the effectiveness estimation (Table 3).

The articles reporting the highest effectiveness fall into the ‘Medium’ to ‘High’ community engagement categories (Figure 2). When comparing articles within the ‘Medium’ to ‘High’ community engagement range, the ‘Medium’ category lacks the consideration of education. Conversely, when comparing the ‘High’ and ‘Medium’ categories with those having ‘Low’ community engagement, the latter do not mention adoption and education as contributing factors (Table 3).

## 4. Discussion

The impact of free-roaming cats on native wildlife and the possible transmission of zoonotic diseases, together with concerns about their welfare and the nuisance they may cause to some members of the community, have been among the main reasons for justifying population control [3,6,44,45]. Various techniques, including TNR methods and their variations (TTVAR and TTVARM), TR, and TE have been employed to control free-roaming cat populations. This study aimed to assess the potential correlation between community members’ level of participation and the effectiveness of these control techniques in reducing the cat population. Among the articles analyzed, only 11 examined the effects of applying control techniques to reduce cat populations over time, and all of them focused on the TNR technique (see Figure 1).

The limited number of articles can be attributed to our specific exclusion criteria, particularly the fourth criterion (see Table 1). While individual sterilization records were commonly documented in these studies (see Appendix A), it appears that temporarily monitoring the population size as a measure of the effectiveness of free-roaming cat control techniques is relatively new (see Figure 1). Therefore, several authors have advocated for experimental research to assess the effectiveness of different control techniques [14,16,45,46].

Regarding the methodology for identifying community member participation, as this is a PRISMA 2020 literature review [26], our study and the types of community member participation focus on what the authors describe in their research, which is also a limitation (see the limitations of the study below).

Implementing population control techniques and monitoring colonies can be costly in terms of money and time [14,47]. Cooperation may be a factor that affects economic cost and time. Thompson et al. [48], in comparing costs of trapping and neutering at a local level (in Knox County, Tennessee), and the procedural and emotional costs of euthanasia, determined that there are two key factors: (1) whether cat caretakers cooperate with management by decreasing the food available to groups of cats and (2) the value attributed to saving the lives of wildlife and free-ranging cats, with TNR being the most economical only if caretakers cooperated. However, if there is no cooperation from cooperators, TE would be the most economically profitable. Despite their costs and the lack of studies evaluating their effectiveness, TNR techniques and their variations are widely popular in various countries, including Chile [49], Italy [35,36], Canada [35], the United States [27,40,42], Japan [50], and Spain [44], among others.

Due to our exclusion of the term “culling” and the pre-established eligibility criteria (see methodology and study limitations), it is not uncommon for all articles to focus on TNR (see Table A1). Another point to note is the low number of articles that met all the criteria, focusing on population reduction, which is an incentive for researchers worldwide to document their efficacy and propose various methodologies for a more concrete evaluation. This assessment should also determine their suitability for the specific context in which they are applied. In cases where these techniques prove ineffective, there may be a need to revise legislation to address not only feline welfare but also to mitigate the adverse impacts of free-roaming cats [44]. Emphasis is also placed on the combination of control techniques, as using a single technique alone may lead to an unintended ‘rebound effect,’ such as recolonization (immigration) in the case of TE [18], or an increase in surrounding cats when applying TNR [51]. Despite focusing on different control techniques, both authors highlight the importance of combining them for long-term effectiveness. For example, TNR could be combined with the euthanasia of sick cats and the adoption of tame cats [52]. However, automated traps can be employed in the case of TE [18].

All the analyzed articles reported varying types of community participation. This may be because most of the reported control applications took place in urban areas, with ten studies conducted in urban settings, and one in a rural area. Although free-roaming cats can have negative environmental impacts on human settlements, their interactions with humans are often positive [7], depending on the area and population where such a technique is applied. Furthermore, all studies exclusively focused on non-lethal techniques. Non-lethal programs, in contrast to lethal alternatives such as TE, often garner positive public perception, making their implementation and community participation more favorable [29,33]. Regarding the type of community engagement, we observed that the primary role is that of ‘caretakers’ (found in 10 studies, see Table 3). This role directly involves the community in the application of the technique and can foster strong bonds between caretakers and colony cats, showing sympathy and ethical concern for their welfare by providing necessary care to the cats [53]. Moreover, cats are more easily trapped by their caretakers than by strangers, which simplifies the implementation of the technique in established cat colonies [47]. When considering alternatives to euthanasia, recruiting caretakers from the local community where the control technique is applied appears to be critically important [52]. On the other hand, having more positive and committed community members also decreases negative publicity and complaints to government officials, possibly increasing funding and assisting in cat population control and ensuring cat welfare [23].

Community engagement through adoption can significantly reduce the cat population, including feral cats, and improve their welfare [39]. While promoting adoption is commonly integrated into the TNR technique [52,53], only seven of the reviewed studies considered adopting feral and free-roaming cats as part of their approach. Notably, these studies demonstrated higher effectiveness than those that omit adoption (Table 3, Figure 2). The exclusion of ‘adoption’ in some studies may introduce bias to the findings, potentially influenced by authors assuming that readers recognize adoption as an integral aspect of trap–neuter–return (TNR).

Education plays a crucial role in shaping cat populations. Natoli et al. [36] analyzed data from colonies registered in Rome between 1991 and 2000, revealing that out of the initial 1665 feral cats, only 1293 remained after a decade. This limited effectiveness can be attributed to the feline immigration caused by abandonment and colony migration. 

Based on these results, Natoli et al. [36] proposed that controlling feral cat populations is ineffective without comprehensive education on the regulation of domestic cat reproduction to prevent abandonment. Examples of science-based community education actions include the following: (1) increasing awareness of feral cat impacts and having trusted foster organizations to take in cats when owners are unable to keep them, along with support groups to provide guidance on cat behavior issues or affordable and accessible medications to discourage abandonment [7,14,16,38]; and (2) providing information on science-based techniques for communities to implement cat population control [27]. Such strategies can be implemented by organizations, authorities, or advocates via community events, media coverage [27,40], and expert involvement, to motivate public action beyond basic leaflets or posters [14,42,43].

There is a close relationship between education, caretakers, and cat welfare [16,53]. As such, the public should be regularly informed of pet ownership responsibilities, including awareness of the effects and implications of neglect. To address ethical issues and ensure the welfare of free-roaming cats, necessary factors include food, shelter, healthcare, and human interaction [16]. Crucially, control techniques should involve collaboration with international and local cat health and welfare agencies. Ideally, their objectives should encompass population control by providing free-roaming animal control facilities and implementing and enforcing policies that ensure ethical control of cat populations for welfare [54].

It must be considered that there are cultural differences in the places where such control techniques were applied, even in the same articles that were applied in different parts of the United States, which does not ensure that the application of such control techniques will have the same effectiveness. To achieve the success of cat population control techniques and the problems they cause in a community, competent local authorities must design and enact laws (for cat welfare and population control) [16], which can be adapted to the context where they will be applied and allow the active participation of the communities [25]. An example of this is provided by Kennedy et al. [7], who considered the best cat population regulation techniques for application in indigenous Australian cultures. There, cats are seen as part of the family and are allowed to roam freely (which in Western society would be considered irresponsible), so the techniques that are commonly applied in Australian indigenous cultures are not very effective. In this research, it is concluded that there is no one-size-fits-all population control plan, and it is necessary to consult and adapt it to the community culturally to achieve a sustainable management program [7].

The most successful application of the control technique, as shown in Figure 2, was observed in studies characterized by a diverse range of community participation types, categorized as ‘High’ participation level, resulting in a reduction in the cat population by over 77%. Spehar and Wolf [27] achieved an extraordinary reduction of 100% (Table 3). In that study, a population control technique was initiated in 1992 in Newburyport, USA. It targeted an initial population of 300 cats and incorporated various strategies, including adoption, focused trapping, and sterilization efforts, the use of keeper feeding records to monitor feline attendance and activity, and educational tactics involving community engagement, media coverage, and informational outreach [27]. This study successfully integrated all four types of engagement mentioned in the results, demonstrating a highly effective approach that resulted in a 100% reduction in the cat population. It should be noted that this study took quite a long time, few resources, and consistent efforts across the period during which the control technique was applied, which is also critical for effective control techniques. It should be noted that a study categorized with ‘Medium’ community participation achieved an effectiveness akin to that of ‘High’ participation studies; however, it required a longer application time (e.g., comparing ID42 with ID40 and ID43, Table 3, Figure 2). This suggests that incorporating additional forms of community engagement, such as education, may enhance the effectiveness within shorter application periods of the control technique.

The level of community engagement and the roles of community members during the application of control techniques are closely tied to the effectiveness of cat population reduction, as indicated by our research. Key roles include adoption and education, which directly influence the number of cats entering and leaving colonies. As proposed by Kilgour et al. [20], strategies for managing free-roaming and feral cats require a comprehensive approach that engages various community aspects and should be conducted over an extended period.

## 5. Study Limitations

Because our methodology is confined to details within the articles, some community participation procedures may have been omitted by the authors. This could have affected the evaluation of effectiveness across participation types. We focused solely on articles documenting the initial and final cat populations, and details on applying and evaluating the control technique. Additionally, we excluded the term “culling” due to its broader implications like shooting and poisoning, which lack humanitarian perception. Consequently, community participation is often overlooked, warranting further research in this area.

## 6. Conclusions

The literature on cat population control has long sparked controversies. This study revealed a significant gap in the assessment of the effectiveness of various control techniques (TNR, TTVARM, TTVAR, TE, and TR). While only covering certain techniques (omitting “culling”), a striking homogeneity emerged in the analyzed articles regarding countries of origin and research authors, with most articles from the United States. Interestingly, different countries have laws that regulate TNR as a method of population control.

Advancing new control techniques or validating existing ones requires further global research combining citizen engagement with experimental trials. Prioritizing citizen engagement is essential because many organizations employ these techniques. For example, sharing results through citizen science initiatives, for example, can enhance the literature on free-roaming cats and control methods. When techniques prove ineffective, legislative revisions should consider both feline welfare and the preservation of native biodiversity.

In conclusion, community engagement played a pivotal role in the application and success of cat population control techniques. Adoption and education are particularly influential, with adoption facilitating rehoming tame cats, and education fostering community awareness about free-roaming and feral cats.

## Figures and Tables

**Figure 1 animals-14-00492-f001:**
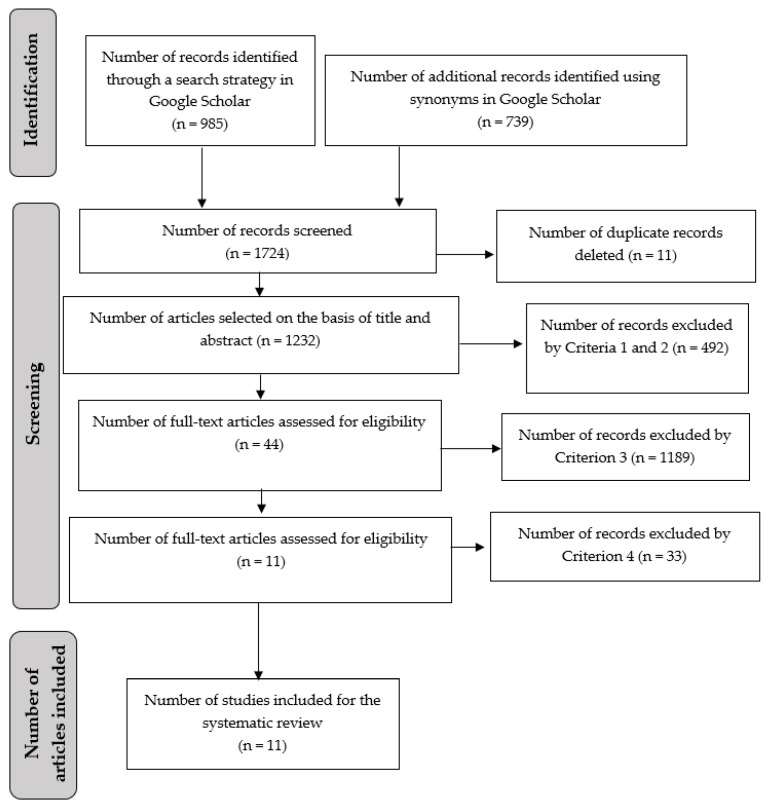
Flowchart illustrating study identification, selection, and inclusion based on the PRISMA guidelines.

**Figure 2 animals-14-00492-f002:**
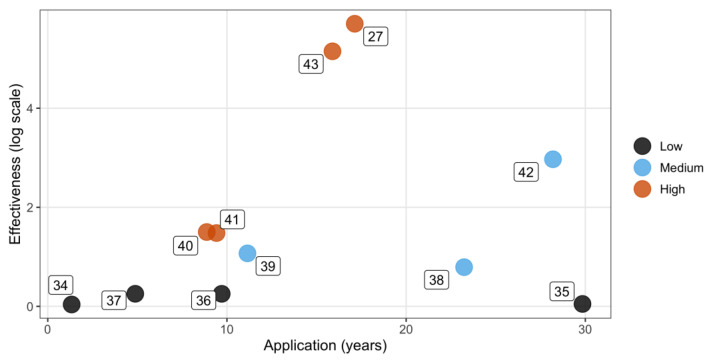
Effectiveness of the control technique in relation to years of application and community engagement level. Colors represent varying levels of community engagement, with numbers corresponding to article identities (ID) (see Appendix A).

**Table 1 animals-14-00492-t001:** Criteria for inclusion in the systematic review on community engagement and effectiveness of free-roaming cat control techniques.

Pre-Established Eligibility Criteria	Description
Criterion 1:	The article should exclusively focus on free-roaming cat populations, including feral cats and owned and semi-owned stray cats.
Criterion 2:	The article addresses the cat population control techniques established in the research objective (TNR and its variations TTVR and TTVRM, TR, or TE). Articles or reports utilizing these techniques in combination were also included.
Criterion 3:	The article focuses on applying a control technique (TNR and its variations TTVR, TTVRM, TR, or TE) within a specific identifiable geographic area (city or country).
Criterion 4:	The article’s objective was to assess a method for reducing free-roaming cat populations by describing the implementation process. This includes the methodology used, initial and final cat population sizes, and the duration of control or evaluation

**Table 2 animals-14-00492-t002:** The type of involvement of the communities during the application of the control technique was identified in 11 articles.

Type of Participation	Description	Number of Studies	Percentage (%)
Caretaker	Caretakers, volunteers from the local communities, oversee and feed the cat colonies where control techniques are employed. They also document population changes in the colonies, including disappearances, emigrations, deaths, and more.	10	91%
Trappers	Volunteers are responsible for trapping cats in the target population where the control technique is to be applied. They also oversee the return of cats in the case of return techniques.	9	82%
Adoption	Community members adopt cats from the controlled population.	7	64%
Education	Community education on responsible cat ownership, cat abandonment, control techniques, and the impacts of cats on native fauna.	4	36%

**Table 3 animals-14-00492-t003:** Information obtained from articles that met the inclusion criteria.

Effectiveness	Initial Population (Pi)	Final Population (Pf)	Type of Participation	Community Engagement	Application	ID
0.03	135	130	Caretakers	Low	1 year	[34]
0.04	15,718	14,973	Caretakers	Low	30 years	[35]
0.25	1665	1293	Caretakers Trappers	Low	10 years	[36]
0.25	62	48	Trappers	Low	5 years	[37]
0.79	455	206	Adoption Caretakers Trappers	Medium	23 years	[38]
1.06	68	23	Adoption Caretakers Trappers	Medium	11 years	[39]
1.48	195	44	Adoption Caretakers Trappers Education	High	9 years	[40]
1.50	69	15	Adoption Caretakers Trappers Education	High	9 years	[41]
2.96	204	10	Adoption Caretakers Trappers	Medium	28 years	[42]
5.14	258	1	Adoption Caretakers Trappers Education	High	16 years	[43]
6.39	300	0	Adoption Caretakers Trappers Education	High	17 years	[27]

## Data Availability

Data are available in Appendix A.

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
