# Peer review of "Community Engagement and the Effectiveness of Free-Roaming Cat Control Techniques: A Systematic Review"

_animals, 2024, doi:10.3390/ani14030492_

Round 1

Reviewer 1 Report

Comments and Suggestions for Authors

Overall, this appears to be a strong paper, and a worthwhile topic.  It is well structured and well written.  In particular, the Methods section is very detailed (appropriately so), and the literature review process was described in a robust and comprehensive manner.  I have recommended a few minor revisions to the manuscript, which are listed below.  The two primary areas for revision are adding greater mention/discussion of how population control techniques relate to cat welfare, and an expanded discussion of potential limitations on the studies reviewed in terms of geographical location and cultural differences.

Simple summary

Page 1, lines 10-11:  it seems including cat welfare, including overpopulation, and transmission of diseases, including zoonotic diseases, would be worthwhile to include here, in order to highlight all of the reasons why cat control techniques are implemented.

Introdution

Page 1, line 41:  should “wildcats” be two words?

Page 1, line 43:  not sure that “Cat War” needs to be capitalized.

Materials and methods

Page 4, line 121:  it would be useful to very briefly explain what the PRISMA guidelines are.

Results

Table 2:  even though it’s mentioned in the text preceding the table, the total number of articles reviewed should be included in the table name.

Table 2:  if the formatting of the article is in United States English, which it seems to be based on word spellings, periods should be used instead of commas in the percentages (as is customary in the US), e.g. 90.91%.

Table 3:  same as above for values in the Effectiveness column

Table 3:  commas should be used instead of periods for the values in the population columns to denote thousands, e.g. 15.718 should be 15,718.

Discussion

Lines 270-271:  it would seem that cats’ welfare should also be mentioned here.  If it has not be addressed in the cited studies, then it could still be included with the caveat that it has not be addressed in research, but is likely an additional reason for population control.

Lines 319-321:  it might be useful to draw a parallel between non-lethal techniques being favorable and the prominent no-kill movement in shelters.

It would be beneficial to briefly discuss practically how communities could be educated, such as who would be doing the educating and by what means the information would be disseminated to the community.  In countries where there might be a bigger problem with disease transmission from cats, which may be in less developed countries, illiteracy in the community could be another potential barrier to education.  (Anecdotally, this seems to be a barrier to involving some communities in addressing free ranging dog issues.)

It would also be beneficial to discuss potential cultural differences between the countries where the 11 studies were conducted in, and how due to that, the findings of these studies might not necessarily be applicable to other countries/regions.  Similarly, depending on where in the United States those studies were conducted, there might also be differences in community attitudes toward cats.

There are a couple of brief mentions of cat welfare throughout, but this seems like an aspect that should be given greater attention, specifically in the Discussion.

Author Response

Dear reviewer:
Thank you very much for your comments that allowed us to enrich this article and improve it, as well as for your patience in reviewing the article. Attached below are the responses to your respective comments, indicating the changes made to the document (with their respective Line) and citation of the modified paragraph.

We will be attentive to any comments and have a happy holiday season.

Reviewer 2 Report

Comments and Suggestions for Authors

Please include the check list from PRISMA as an appendix.

I really am excited to see the community context evaluated in the context of free-roaming cat population control. This has been completely absent. And a good systematic review is a fabulous way to assess that. I do have some concerns and suggestions about how the paper is framed in the introduction, the search method, and how the criteria for inclusion were applied. See below for specifics.

My first concern is how the introduction is framed, specifically lines 47 and 54 onward. These statements don’t really contribute to the main point of this systematic review. There are many additional reasons why humans may want to manage/decrease free-roaming cat populations beside potential predation. If the purpose is to understand community collaboration and socio-environmental context (an exciting lens for this systematic review) and how it impacts population decline, then that should be more the focus of the introduction. How many animals cats kill isn’t really relevant—and the importance of free-roaming cat population control can be justified in many ways beyond that measurement (which is often extrapolated inappropriately). Please edit so that the need for control is briefly stated and documented broadly (including cat welfare and nuisance issues).  If the authors feel additional justification beyond referencing some recent reviews are needed, please include all the reasons for population management. Giving more space to the polarization of the conflict only inhibits discussion and working together for solutions. See also lines 270-3: in the discussion. Please edit.

Many of the definitions are not the most widely used ones (there are several options but still some commonly used versions). Please use definitions that clearly separate the different groups of cats and illustrate how the populations can be related (and difficult to correctly identify).

The critical importance of community engagement in managing cats should be better emphasized. If there are other parallel examples that would be a crucial piece to include and describe in the introduction.

My second major concern is that at least a few references that I think should have been found and included in the full review didn’t appear in the search strategy. I’ve listed them below and included my recommendations to address this because I suspect that other references, perhaps from the wildlife side, are missing. I see the authors’ decision about excluding culling and other less humane methods, but if the intent was to see how effective different methods were relative to their community collaboration that is a potentially large segment of work. Not that I like these rather inhumane methods but if they are to be excluded, then the full range of potential control methods are not going to be represented here and much of the text and context will need to be edited to be clear these other options were deliberately excluded.

Abstract: line 3-33: also, limitation by only ability to crudely estimate community engagement.

Line 43: polarizing the perspectives this way is not constructive to achieving a dialogue or solutions so I would recommend removing this last phrase.

Lines 52-53: I don’t think these adjective “aggressive” is useful here. And any free roaming cat can be aggressive in predation or carry disease…it just depends on the individual cat. Are the definitions above from the references or a mix of those plus the authors’ definitions? Please clarify or simplify, particularly as any unowned free-roaming cat is really the focus of this work.

Line 54-5: Feral cats are by definition free-roaming unless they are in a shelter or sanctuary. The free-roaming population is a mix of owned and unowned, more and less socialized. What is the purpose of this paragraph? Line 57: any cat could carry toxoplasmosis if they ate infected prey.

Line 70-1 and following: trap-neuter-return is the more correct terminology. And it does not include owned cats in most locations because the laws are different for owned cats vs unowned cats. Only unowned, free-roaming cats are typically included in TNR programs (usually there are some assumptions to separate owned from unowned like a collar or being very friendly or telling cat owners to keep their cats indoors during trapping…). The distinction and definitions are not quite accurate and aren’t really needed. There are many variations beyond the basics of trapping, neutering, and returning to where the cats came from. The majority of TNR programs include ear-tipping, and many include vaccinations. Fewer include testing for FeLV or FIV in recent years. Some microchip the cats. Some work with caretakers and some just do the TNR services. A few include monitoring. Please find a recent reference and describe the variations as they are important for this study.

Line 77-84: Again, these definition are not quite correct. Trap and Remove may be for adoption or for euthanasia. So that (4) is a subset of (3). And (4) is often NOT performed in any organized way by animal shelters. Shelters may address nuisance complaints by trapping cats; those cats may then be adopted or euthanized. And I’m not sure that TE is less controversial but perhaps is more acceptable than the description in the next paragraph of other lethal options.  Lethal options should be grouped together.

Line 87: this is an overstatement. While there are islands where cats have been eradicated, this is relatively rare and very costly. Where cats have not been eradicated, ongoing lethal control is needed to attempt to keep the cat population under control. Furthermore, in these situations, the benefit to the native species is varied. See for example Bergstrom et al J Applied Ecology 2009 and Scomparin et al Biol Invasions 2023. 

Line 95: success of a program can be a hot button because the definition is quite variable. Even as stated here: “success in controlling cat populations” isn’t quite clear. Does controlling mean stabilizing the numbers or ages? Or is it reducing or even eliminating the population? What is the goal of the TNR effort (as the authors have included in their criteria)? Also as noted, what is the timeline? No one would argue that TNR will take time before the numbers of cats decline. Please reword for clarity.

Line 146 and following: Looking at the summaries (and in some cases the original article) Ids 13, 17, 18, 19, 22, 26, 29, 33, 36 seem like they should be included in the A3 table. I would like to see the specific reason articles were excluded from A3 added to Table A2. There are also several articles missing from A2 including Castillo and Clark 2003 Natural Areas Journal; two in JAVMA: Andersen 2004 and McCarthy 2013; Gunther et al PNAS 2022; and Tan et al Animals 2017. Tan came up in google scholar, but the others were not there. There may be some additional articles missing. Please consider 1. Using additional search engines and 2. Reviewing references from existing articles for completeness of the list.

Cell D42 A2 has a formatting error.

Line 175: why was this chosen for identification of community engagement? This is a pretty crude measurement. Please explain more about what options were considered but were not possible here. And include some additional recommendations for how to document community participation in the discussion. Were there any animal shelter or rescue groups engaged? If so, how were they classified below?

Thank you for figure 1!

Table 2 and throughout. Please round percentages to whole numbers for >=10% and 1 decimal for < 10%. This makes it easier to understand the relative differences between percentages when reading the article.

Line 234-5: please give an example of this calculation here for less sophisticated readers. Would the percent change work as well? That is a much easier to grasp type of number.

Table 3: if effectiveness is a crucial variable (I’m assuming this because it is the sorting variable, then put that in the first column of the table and put the ID last (so just switch effectiveness and ID). And order from most to least effective.

Love figure 2: to what do you attribute the low effectively of ids 41 and 40 given the high involvement? That would seem to counterbalance the effectiveness and high engagement findings for ids 27 and 43. And contradict the statements about increasing effectiveness and high engagement in the manuscript. Please address.

Line 285: the importance of monitoring populations and some simpler methods to do it in a reliable way is relatively new. For your information, the DC Cat Count validated transect counts (and see the publications which are still coming out) as a method for cats. And the Alliance for Contraception in Cats and Dogs has put together a recent document to explain other simple monitoring options.

Line 292: this reference by Lohr includes costs for bird deaths which are highly variable. If the authors wish to discuss costs, they should include all the simulations which examine those costs, including the cost of kitten deaths (see Boone et al Frontiers in Vet Sci 2019 and Benka et al J Feline Med Surg 2021 as two examples). But I’m not sure that this topic really adds to the discussion about community engagement. The Thompson et al article (id #9) modeled population size and what might happen if caretakers decreased feeding as population size declined. Perhaps that is a better example of how community engagement could be helpful?

Line 295: “approved” is a strong word. At least in North America, TNR is variable practiced as it is often illegal under some of the old laws about abandonment. And in some locations, it is illegal to feed cats. Please edit this statement.

Line 318: “often positive” I think that this is a bit overstated because there are certainly some citizens and residents who are very angry about having cats roaming in their neighborhood or yard or scratching their cars or urinating on their front doors.  On the other side, an article just came out (Crawford et al Animals 2023 (in October) about caretakers and TNR in Australia showing how TNR improves the caretakers’ lives. Please add a caveat that this “positive” is somewhat variable depending on the location and population.

Line 330-1: adoption is often considered by wildlife researchers to NOT be part of TNR (it isn’t in the abbreviation) so please edit this statement. Adoption is very often practiced as part of TNR but needs to be explicitly stated as part of the work.

Line 359: and this took quite a long time, quite a few resources, and consistent effort across the time period—that consistent long-term effort is also critical. Please add this information here.

Please add the limitations of this study.

Comments on the Quality of English Language

In a number of places there are very minor edits to English that would make the grammar more correct. However, the manuscript is fully understandable.

Author Response

(The authors gave the same response as above.)

Reviewer 3 Report

Comments and Suggestions for Authors

Author Response

(The authors gave the same response as above.)

Round 2

Reviewer 2 Report

Comments and Suggestions for Authors

Thank you for your responses and edits. The manuscript is much improved. In some places the authors haven't done quite enough to completely address my comments. See below.

Since culling isn’t included, modify summary and abstract to reflect that only a subset of techniques was included in the review.

Line 19: there are quite a few publications on the control of cat populations, but few include or evaluate community engagement…nor do many include all of the components that were required for this review. Please edit language here and throughout to reflect this (as was done in the abstract for example).

Line 53-4: this isn’t a complete definition from that article. And that article creates a rather artificial division between feral and unowned free-roaming cats because it is based on Australian law. Feral cats are free-roaming. Please either use a different reference or completely include the information from this reference. For example, free-roaming cats do include owned cats, they just roam some or all of the time. The simpler version of this set of definitions is also used in Kennedy’s review (reference 7). And Line 75 (and others) uses stray cats…still not defined in the manuscript.

Line 76-8: Cats are trapped, and sometimes tested for FeLV and FIV, vaccinated against rabies, have their ears cleaned, sterilized, and then re-leased at the capture site.  Please edit to “Cats are trapped, vaccinated against rabies where that disease occurs, ear-tipped (1 cm of the left ear removed to indicate that they are sterilized), sterilized, and returned to their capture site.”

Line 78-9: first variation includes testing and may or may not explicitly include a caretaker. And uses Altered instead of neutered. The second variation can be omitted or explained more clearly.  Please edit.

Line 83: TR can also include euthanasia of any cat which is trapped. And may include placing cats in sanctuaries instead of adoption or euthanasia. Please edit.

Line 101: I understand what the response to reviewers is, but many organizations will argue that the success of their program is based on the # of cats sterilized or litters prevented or…Please explicitly define what the authors mean as “success of control programs”. Many programs do NOT have stabilizing or decreasing cat populations as a goal.

Line 112: it isn’t clear with the edits what the “its” is referring to. Please edit for clarity.

PRISMA guidelines: thank you for adding. Is there a reason that items 14 and 15 weren’t performed? Seems like they should have been or were in fact included in the assessment of the articles based on the criteria used for inclusions? Similarly, 16b, this was done as part of the assessment, wasn’t it? Also, for many of the items, the “no” is actually “not applicable” because a statistical synthesis wasn’t used. “N/A” might be a better response for those items for clarity.

Table A3: the listing from the response to reviewers starting on the bottom of page 5 should be added to New Table A3 so that the reader understands the rationale for exclusion without having to read each article. Please add a brief version to applicable rows of that table.

Line 139: this definition isn’t the same as the one in the introduction. Please edit. The reader needs to see this here in addition to the limitations section as this is a bias some will find to be objectionable. And this statement in the response to reviewers is important to include here in some form:  As a result, our focus is on established techniques that can be applied across different contexts without generating substantial controversy while ensuring the welfare of the cats as much as possible.

Table 2: My request was to keep larger numbers whole with rounding and only include a decimal for smaller numbers in tables and text: As a result, our focus is on established techniques that can be applied across different contexts without generating substantial controversy while ensuring the welfare of the cats as much as possible. It is just too difficult to read easily otherwise. And title needs to be standalone—that this is a systematic review of….

Discussion: 42 Medium level of engagement but high effectiveness? Please discuss in the discussion why this might be the case. And therefore, is an exception to the overall conclusions.

Line 288: Alliance for Contraception isn’t working with DC Cat Count; they are using DC Cat Count information to create new materials.

Line 296: please note that this is a basic model of one location which reached these conclusions.

Line 306: and culling was excluded which is mostly the term that wildlife protection approaches use. Please add. The review was therefore biased against these types of studies. And why my comment about line 139 is critically important to address here as well.

Line 330: what does this review have to do with “no-kill”? Either remove or make clear in the text.

Line 335-6: also having more positive and engaged community members decreases negative publicity and complaints to government officials. Fewer complaints are likely to increase funding and other support and limit punitive laws being passed. Please add this context here.

Line 343: this lack of considering “adoption” as part of the approach is also likely due to the authors assuming that the readers knew that was going to happen as part of TNR. Please add that here as this is a source of bias in the review.

Line 354: what is responsible pet ownership? Please avoid that term and include the type of care which it represents for the authors. What are other ways to discourage abandonment? How about: Having a trusted shelter organization to take cats when owners can’t keep them. Knowing where to go for help with behavior or medical issues that are affordable and accessible. This is an important concluding point so please include more aspects of this topic. It isn’t a simple solution.

Lines 358-9: and all of this information has to be presented in a way that is understandable by the audience. Plus, it should rely on underlying theories to change human behavior. Again, this isn’t just putting together a brochure but rather engaging experts to create compelling motivation and options for the public to take action. Please expand this section a bit here and below.

Line 367-8: unfortunately, agencies often do NOT have population control as one of their missions. Rather, they are either punitive and enforcement based using old and outdated laws which don’t apply to free-roaming cats, or they are public health and regulatory based. I’d love for this to always be true, but it isn’t. Please edit and include the idea that this SHOULD be the situation and isn’t always.

Line 370 and following: great addition and really important!

Line 425: there is homogeneity IN THIS REVIEW because culling was excluded. Please edit.

Comments on the Quality of English Language

In some places, the important points that the authors are trying to make are made a bit less clear by the language limitations. I would really like to see this edited and improved so that the full impact of the publication will be easily accessible for all readers.

Author Response

Thank you very much for thoroughly reviewing our manuscript. We apologize for not completely addressing all your previous comments. In this revised version, we have carefully addressed your feedback.

Round 3

Reviewer 2 Report

Comments and Suggestions for Authors

Thank you for your attention to detail and patient improvement of the manuscript.

Comments on the Quality of English Language

I believe that there should be some editing of English for professional standards, especially for non-native English speakers trying to read the manuscript.